# Evaluating the validity and reliability of the Tswana adaptation of the MOS-HIV tool for health-related quality of life among HIV Sub-populations in Botswana: A study protocol

Mooketsi Molefi[1]*, Olanrewaju Oladimeji[1,2]

1 Department of Family Medicine & Public Health Medicine, Public Health Medicine Unit, Faculty of Medicine, University of Botswana, Gaborone, Botswana, 2 Department of Epidemiology and Biostatistics, School of Public Health, Sefako Makgatho Health Sciences University, Pretoria, South Africa

* molefim@ub.ac.bw

**Data Availability Statement:** Deidentified research data will be made publicly available when the study is completed and published.

## Abstract

The lack of culturally and contextually appropriate adaptations of health-related quality of life (HRQoL) tools hinders HIV patient outcomes. This study aims to assess the validity and reliability of a Tswana version of the Medical Outcome Survey-HIV (MOS-HIV) tool among diverse HIV sub-populations in Botswana. In terms of the methodology the study will comprise of several steps. Firstly, forward and back-translation of the original U.S. English MOS-HIV tool into Setswana, followed by the review of the translated tool. Phase 1 will evaluate content, construct validity, and reliability of the newly developed tool among HIV outpatients at Gaborone Infectious Diseases Clinics (IDCC). A Cronbach's alpha coefficient >0.7 across the 35 items and 11 dimensions of the MOS-HIV tool will indicate internal consistency reliability. Phase 2 will employ the use of logistic regression models to identify predictors of poor HRQoL among randomly selected IDCC sites, both in urban and rural centers. Phase 3 will investigate predictors of poor HRQoL among inpatients receiving treatment for HIV-associated cryptococcal meningitis (CM) using longitudinal data analysis methods. Ethical approval has been obtained from the University of Botswana, Walter Sisulu University, Human Research and Development Unit, Ministry of Health, and Princess Marina Hospital. Prospective participants will provide written informed consent, with proxy consent explored when feasible. Voluntary participation and confidentiality will be ensured during data collection and analysis. Data will be securely stored under lock-and-key. Dissemination of study findings will adhere to strict privacy protocols, avoiding the sharing of personal identifiers.

## Introduction

The current understanding of patient well-being during medical treatment primarily focuses on the improvement of laboratory and clinical parameters, often overlooking patients' subjective experiences and perceptions. In many developing regions, including sub-Saharan Africa,

**Funding:** The author(s) received no specific funding for this work.

little attention is paid to patients' views and feelings, which are essential for successful recovery. Moreover, patients are frequently excluded from the decision-making process, despite evidence suggesting that their involvement leads to cost-effective services and improved outcomes [1–3].

Compounding this issue is the lack of appropriate tools for objectively assessing patients' quality of life (QoL) during treatment. These tools, where available, significantly contribute to better patient outcomes by providing additional non-clinical information to guide clinical providers in patient care [4]. Specifically, for HIV/AIDS patients, who form a significant proportion of the population in sub-Saharan Africa, there is a scarcity of validated tools for assessing health-related quality of life (HRQoL) tailored to this setting. HIV/AIDS, especially in advanced stages, can be debilitating, akin to other chronic conditions like cancer and diabetes mellitus [5].

The Medical Outcomes Study HIV Health Survey (MOS-HIV) tool has emerged as a comprehensive instrument for assessing HRQoL among people living with HIV/AIDS (PLWHA) [6]. Initially developed by Albert Wu, the MOS-HIV tool has been refined over time to cover various dimensions of HRQoL, making it a widely accepted measure in HIV populations across different settings [7]. Despite its success, translations of the MOS-HIV tool into local languages in Africa are limited, despite the region bearing the highest burden of HIV/AIDS globally [8, 9]

In response to the gaps in patient-centred care and HRQoL assessment in HIV/AIDS management, this study proposes the development and evaluation of a culturally relevant version of the MOS-HIV tool in Botswana, a country heavily affected by the HIV epidemic. Despite Botswana's commendable efforts in antiretroviral therapy (ART) roll-out [10], holistic care for PLWHA remains inadequate, with minimal attention given to psychosocial well-being.

The MOS-HIV tool comprises ten dimensions covering various aspects of HRQoL, including physical, role, social, and cognitive functioning, as well as pain, mental health, distress, energy, quality of life, and general health perception [6, 11]. Originally derived from the MOS short-form, the MOS-HIV tool has undergone continuous refinement to capture a broader range of HRQoL dimensions [12].

## Reliability

Previous studies have consistently reported high internal consistency for the MOS-HIV tool across different HIV populations, with Cronbach's alpha exceeding 0.7 [6]. The tool has demonstrated satisfactory reliability across all disease stages and CD4 categories [13, 14].

## Construct validity

Numerous studies have affirmed the construct validity of the MOS-HIV tool, showing high convergent and discriminant validity across various disease stages and health indicators [15, 16]. Scores on the MOS-HIV tool have been found to correlate with other established health instruments, indicating its robust construct validity [17].

## Responsiveness

The MOS-HIV tool has shown sufficient responsiveness to clinical changes in HIV populations, with scores correlating with improvements or declines in health status [18]. However, its responsiveness to specific conditions like cryptococcal meningitis remains unexplored, highlighting a gap in research.

### Local adaptation and versions

While the MOS-HIV tool has been translated into several languages globally, its adoption in African countries, particularly sub-Saharan Africa, has been limited. Only a few studies have attempted translation and validation in African settings, emphasizing the need for culturally relevant tools to assess HRQoL among PLWHA.

### HRQoL in the HAART Era

Despite significant advancements in HIV treatment, HRQoL assessment among PLWHA in sub-Saharan Africa remains understudied [19]. Late presentation to medical care and the prevalence of life-threatening conditions like cryptococcal meningitis underscore the importance of evaluating HRQoL during treatment.

### HRQoL among patients with advanced HIV

Studies evaluating HRQoL among PLWHA with advanced disease in sub-Saharan Africa are scarce. Understanding HRQoL changes during treatment, particularly for conditions like cryptococcal meningitis, is crucial for improving patient care and outcomes [20–22].

The proposed study aims to address the gaps in HRQoL assessment among PLWHA in Botswana by developing and evaluating a culturally relevant version of the MOS-HIV tool. By validating this tool in a local context and identifying predictors of poor HRQoL, the study seeks to enhance patient-centered care and inform policy changes. Additionally, the study aims to contribute to the broader understanding of HRQoL among PLWHA in sub-Saharan Africa, potentially influencing future research and interventions in the region.

## Methods

### Study design and settings

This study will be conducted in three phases, following a translation process that involves forward and back-translation for accuracy. A pilot testing study will be used to determine the content and construct validity of the adapted tool among patients accessing the HIV outpatient clinic at Princess Marina Hospital, specifically the Infectious Diseases Clinical Care Center (IDCC). Princess Marina Hospital is a 592 bed capacity tertiary hospital situated in the centre of the capital city Gaborone [23]. The reliability of the measurement tool will be assessed using inter-rater reliability and test-retest reliability [24]. To identify predictors of poor HRQoL, patients accessing the IDCC will be enrolled from four randomly selected urban and rural settings. Phase 3 will evaluate the Tswana MOS-HIV in an inpatient setting using a prospective cohort study design, tracking changes in HRQoL over time among men and women admitted for AIDS-related cryptococcal meningitis.

### Translation

After permission is granted by MAPI Trust® to translate the tool, the translation process will begin with the involvement of Tswana-English language experts. At least two independent linguists in the forward translation of the tool, and the other pair for the back-translation. At each of the steps, one of these experts will be blinded from the concepts that are to be studied. Any discrepancies between the two versions will be addressed by reaching a consensus among the research team and the translators.

## Preliminary pilot testing

This process will take place between the translation process by an expert committee and Phase I of the study. The new Tswana tool will be pre-tested among 30 HIV patients attending the Village clinic IDCC in Gaborone. The data collected here will not form part of the analysis in the main study but will help in several ways. It shall help detect how long the respondents take to complete the survey and whether they understand the questions. Through this process, potential distribution of scores in the main study could be appreciated. Pre-testing will reveal any deficiencies such as floor and ceiling effects of some items, feasibility of undertaking the project and potential review of question order post-validation.

## Validation process

The newly adapted tool will be evaluated for content validity(including face validity) assessing to what extent all the items of the construct, all together, comprehensively reflect the construct to be measured, considering even the target population. The structural validity will be carried out via exploratory factor analysis. Internal consistency will be judged by the presence of limited unidimensionality and Cronbach's alpha, while reliability will be assessed via a weighted kappa.

## Sampling procedure

For phase 1, systematic sampling will be employed as patients are seen in a queue at the health facility. The sampling frame will be drawn from the total number of registered patients due for a visit during data collection. The interval (K) will be determined based on the desired sample size. In phase 2, stratified random sampling followed by multistage sampling will be used. Sites will be stratified by region (north or south) and administrative classification (rural or urban), and one site from each stratum will be selected for systematic sampling selection. In phase 3, all eligible prospective participants with cryptococcal meningitis will be screened and enrolled without sampling.

## Sample size determination

The minimum sample size required for validation of the newly translated tool will be determined using the formula recommended by Bonnet et al [25]. Assumptions include a desired Cronbach's alpha of ≥0.7 [24], an effect size of 1.2, and known values for c and δ. Adjusting for a response rate of 80% and a respondent-to-item ratio of 5:1, the sample size for phase 1 will be 220 participants. For phase 2, sample sizes per site will be determined based on a conservative prevalence of 50% (rural) and 25% (urban). The resulting sample sizes are 420 and 300 for urban and rural areas, respectively. For phase 3, a minimum sample size of 36 is calculated to detect the largest changes in mean scores per week using repeated measures ANOVA. With adjustments for systematic error, the final sample size for phase 3 will be 40.

## Inclusion/exclusion criteria

Phase 1 includes HIV patients on antiretroviral therapy (ART), regardless of CD4 count, and those registered at IDCC. Exclusions include HIV-negative patients, those with unknown HIV status, patients accessing care elsewhere, and minors (age <18 years). Phase 2 aims to enrol HIV patients from four randomly selected IDCC sites, excluding those in admission or receiving care from other sites, minors, and individuals unable to provide consent. Phase 3 includes CM patients admitted for treatment, excluding minors, individuals unable to provide informed consent, and those with >48 hours of cryptococcal meningitis.

## Study procedures

The study will proceed sequentially through the phases. Permission to translate the tool has been obtained, and independent language experts will perform forward and back-translation. Research assistants will be hired and trained for data collection and entry. Pre-testing of the tool will be conducted with 30 HIV patients attending the Village clinic IDCC in Gaborone. Data will be stored in Redcap software and analyzed using Stata. Reliability will be assessed using intra-class coefficient and agreement analyzed using Cohen's kappa. Demographic summaries will be provided for phase 1. Validity will be measured qualitatively for face and content, and construct validity will be assessed using confirmatory factor analysis. Binary logistic regression will explore associations between demographic and clinical variables and poor HRQoL in phase 2. ANOVA or nonparametric tests will detect differences and changes in HRQoL scores during phase 3.

## Ethical considerations

Ethics approval has been obtained from the Ministry of Health Human Research and Development Committee. The study has been approved by the University of Botswana IRB reference number UBR/RES/IRB/BIO/278, Ministry of Health's Human Research & development committee, permit reference number HDPME 6/14/1;Princess Marina Hospital IRBPMH 2/2A(7)/230 and the Walter Sisulu University IRB permit number 034/2021. Data privacy will be ensured with unique study identity numbers and restricted access. Patient confidentiality will be maintained throughout the study, and ethical principles of respect for persons, beneficence, and autonomy will be observed during the consenting process.

## Data analysis

Data will be stored in Redcap software [26] and transferred to Stata version 18 [27] for analysis. Reliability will be assessed using the intra-class coefficient and agreement will be measured using Cohen's kappa analysis. Cronbach's alpha will be reported as a measure of internal consistency. A Cronbach's alpha of $\geq 0.7$ in 80% of the sub-scales will be considered sufficient. Face and content validity will be evaluated qualitatively, while construct validity will be assessed using confirmatory factor analysis. In phase 2, the results will be categorized as "poor HRQoL" or "good HRQoL" based on MOS-HIV summary scores below or above 50%, respectively. Binary logistic regression will be used to explore the association between demographic variables (e.g., age, sex, marital status) and poor HRQoL. Clinical factors such as CD4 count, ART regimen, adherence levels, and co-morbidities will also be evaluated using the same approach. Bivariate analysis will be conducted initially, followed by multivariable regression modelling for variables with a p-value<0.25. A backward hierarchical modelling approach will be employed to develop a concise prediction model. In phase 3, mean or median summary scores will be calculated based on patients' follow-up schedule. One-way ANOVA or Kruskal-Wallis ANOVA will be used to detect differences in overall HRQoL scores by visit, while repeated measures ANOVA or Friedman test will be used to identify changes in HRQoL scores over the study period. Odds ratio will be used to quantify associations, and statistical significance and measurement precision will be reported using p-value<0.05 and 95% CI, respectively.

## Discussion

The development of a Tswana version of the Medical Outcome Survey of HIV tool (MOS-HIV) and assessing its reliability and validity among HIV sub-populations, both out-

patients and in-patients receiving treatment for cryptococcal meningitis (CM), holds significant benefits and value for several reasons. This section will highlight these advantages and compare the likely outcomes to those observed in other settings where the MOS-HIV tool has been utilized.

Firstly, the adaptation of the MOS-HIV tool to the Tswana language allows for a culturally and linguistically appropriate assessment of health-related quality of life (HRQoL) among individuals living with HIV in Botswana. By translating the tool, we ensure that the questions and response options are easily understood and reflect the experiences and perspectives of the local population. This localization enhances the tool's relevance and increases its applicability in the Botswana context.

The assessment of reliability and validity of the Tswana MOS-HIV tool among both out-patients and in-patients receiving treatment for CM provides a comprehensive understanding of its performance across different clinical settings. This evaluation enables us to determine whether the tool consistently measures HRQoL and produces reliable results among these distinct HIV sub-populations. It is crucial to ascertain that the tool demonstrates good internal consistency, test-retest reliability, and agreement between different language versions to ensure its robustness and usefulness in clinical and research settings.

By comparing the likely outcomes of the Tswana MOS-HIV tool to those observed in other settings where the tool has been employed, we can assess the generalizability and consistency of HRQoL measurements across diverse populations. This comparative analysis allows us to identify similarities and differences in HRQoL profiles among individuals living with HIV, both within and outside of Botswana. Understanding these patterns can contribute to the global knowledge base on HRQoL outcomes in HIV populations and provide valuable insights for cross-cultural comparisons and interventions.

Additionally, the use of the MOS-HIV tool in both out-patient and in-patient settings allows for a comprehensive assessment of HRQoL throughout the continuum of care. Evaluating HRQoL among out-patients accessing HIV clinical care at the Princess Marina Hospital's Infectious Diseases Clinical Care Center (IDCC) provides insights into the long-term impact of HIV on quality of life in the community. On the other hand, examining HRQoL among in-patients receiving treatment for CM gives us a unique perspective on the challenges and burden of a specific HIV-related condition. This comprehensive approach helps us understand the complex interplay between disease management, treatment outcomes, and HRQoL in different care settings.

Furthermore, the findings from the validation of the Tswana MOS-HIV tool in Botswana can inform healthcare providers, policymakers, and researchers about the specific HRQoL needs and priorities of individuals living with HIV in the country. The results can guide the development and implementation of targeted interventions and support services tailored to address the identified challenges and promote better HRQoL outcomes. By understanding the factors influencing HRQoL, strategies can be devised to improve treatment adherence, social support systems, and mental health support, ultimately enhancing overall well-being in this population.

It is worth noting that comparing the likely outcomes of the Tswana MOS-HIV tool with studies conducted in other settings might reveal variations attributable to cultural, socio-economic, and healthcare system differences. These comparisons will enable us to identify potential contextual factors that influence HRQoL outcomes, providing insights into the unique challenges and strengths of the Botswana HIV care landscape.

This study has notable strengths and limitations. By adapting the MOS-HIV tool into Tswana, the study shows sensitivity to cultural and linguistic nuances, potentially enhancing participant understanding and accuracy of response. The study assesses the Health-Related

Quality of Life (HRQoL) in various HIV subpopulations, offering a wide range of insights that can be applied to diverse groups within Botswana's HIV patient community. Additionally, the study seeks not only to apply the Tswana version of the MOS-HIV tool, but also to rigorously evaluate its reliability and validity. This rigorous testing will ensure the tool's effectiveness and scientific integrity. While the tool's adaptation to Tswana language and context is a strength, it may limit the study's applicability in other linguistic or cultural settings, reducing the generalizability of the results. We note, the reliance on the MOS-HIV tool for assessment could potentially be viewed as a limitation, and perhaps arguments for a multi-method approach could be made; however, given that the MOS-HIV tool has demonstrated high reliability and validity across different settings, we do not anticipate any sub-performance in our setting.

Lastly, factors such as socioeconomic status, literacy levels, or stigma around HIV, that can also significantly influence HRQoL, may not be directly addressed or controlled in this study, possibly leading to limited conclusions.

In conclusion, the development and validation of the Tswana MOS-HIV tool among HIV sub-populations in Botswana offer significant benefits and values. This research contributes to the localized understanding of HRQoL in individuals.

## Author Contributions

**Conceptualization:** Mooketsi Molefi.

**Supervision:** Olanrewaju Oladimeji.

**Writing – original draft:** Mooketsi Molefi.

**Writing – review & editing:** Mooketsi Molefi.

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
