## [Decision Letter · Decision Letter 0]

18 Dec 2023

PONE-D-23-31341Evaluating the validity and reliability of the Tswana adaptation of the Medical Outcome survey tool for health-related Quality of life among HIV sub-populations in Botswana: A study protocolPLOS ONE

Dear Dr. Molefi,

Thank you for submitting your manuscript to PLOS ONE. After careful consideration, we feel that it has merit but does not fully meet PLOS ONE’s publication criteria as it currently stands. Therefore, we invite you to submit a revised version of the manuscript that addresses the points raised during the review process.

We look forward to receiving your revised manuscript.

Kind regards,

Richard Makurumidze

Academic Editor

PLOS ONE

“The project will receive support from the University of Botswana's Office of Staff Development & Training. Additionally, publications and relevant workshops, including capacity enhancement training, will be subsidized by supervisor grants: National Research Foundation (NRF) Grant No. 132385 (Incentive Funding for Rated Researchers - IPRR) and Award Number D43 TW010543 from the Fogarty International Center and National Institute of Mental Health, of the National Institutes of Health.”

3. Please amend either the title on the online submission form (via Edit Submission) or the title in the manuscript so that they are identical

Reviewers' comments:

Reviewer's Responses to Questions

**Comments to the Author**

1. Does the manuscript provide a valid rationale for the proposed study, with clearly identified and justified research questions?

Reviewer #1: Partly

Reviewer #2: Yes

Reviewer #3: Partly

2. Is the protocol technically sound and planned in a manner that will lead to a meaningful outcome and allow testing the stated hypotheses?

Reviewer #1: No

Reviewer #2: Yes

Reviewer #3: Partly

3. Is the methodology feasible and described in sufficient detail to allow the work to be replicable?

Reviewer #1: No

Reviewer #2: Yes

Reviewer #3: No

4. Have the authors described where all data underlying the findings will be made available when the study is complete?

Reviewer #1: No

Reviewer #2: Yes

Reviewer #3: Yes

5. Is the manuscript presented in an intelligible fashion and written in standard English?

Reviewer #1: Yes

Reviewer #2: Yes

Reviewer #3: No

6. Review Comments to the Author

You may also provide optional suggestions and comments to authors that they might find helpful in planning their study.

Reviewer #1: Dear authors,

Thank you for submitting your research protocol for your study to PlosONE. Unfortunately, I found this manuscript to not meet the minimum quality standard for a study protocol. I can only advise to

- consider using guidelines, e.g. the COSMIN guidelines for the development and validation of patient-reported outcome measures under www.cosmin.nl

- consider accessing textbooks on the subject matter, whether by the COSMIN group (see "Measurement in Medicine" by Henrica De Vet and co-authors, 2011) or even good review articles in seminal journal (i.e., the Lohr/ISPOR guidelines),

- consider using guidelines by the MAPI trust for the translation of the tool to your language/culture

Just a few pointers:

- Cronbach's alpha is NOT reliability. For reliability, you want to assess inter-rater or test-retest reliability or both. Both should be done using ICCs or kappa coefficients.

- Cronbach's alpha without a hint as to the factor structure of the tool is unintelligible. Also, Cronbach's alpha is not defined for multidimensional scales (see Tenko Raykov's work). When adapting a well-known instrument, you at least want to confirm its factor structure, ideally using measurement invariance techniques.

- Logistic regression analysis in a cross-sectional sample with an ill-specified sampling mechanism (what you do is not "systematic random sampling", by the way: "systematic random sampling" does not exist - please look into any textbook on sampling mechanisms) and formulating this as "predictors" is not an appropriate technique for a validation study. Please reconsider the definition and subtypes of construct validity, see the COSMIN guidelines on how this should be accomplished.

- Presenting a study protocol for a validation study without describing the tool or the validation studies on the tool that have gone before is a feat in itself! I think I have never come across a validation study or protocol that has not described what the tool is actually measuring. You also would want to give a history of its validation journey and - in particular - where you see the main cultural issues when translating into sub-Saharan languages/culture.

- For this part in particular, you would want to consider using cognitive interviewing techniques and most likely quite extensive qualitative techniques to make sure all items, the construct, and the measurement scales are understood the same or need adapting in your culture.

As inspiration, I can highly recommend Eve Namisango's work from Uganda - please have a look into how she accomplished translation work in palliative care.

Reviewer #2: This is a much needed study within the geographical area of Southern Africa where test translation and validation studies are not as prevalent. Much research is needed on psychological measures, as well as the neglected area of the well-being of those infected with HIV.

The student and their promotor should be commended on this exciting proposed study.

The document reads very well, and all the important points and considerations are covered, in my opinion. The projected sample sizes are also good.

All I would suggest is that they add the full stops at the end of sentences (eg. lines 94, 161, 181, 253, 256, and 260). Apart from that I wouldn't change anything, as the protocol is specific enough, but allows adequate room for statistical exploration.

Reviewer #3: Preamble

The paper that has been submitted ia a protocol for a study to assess validity and reliability of a widely used tool taht assesses quality of life among peple living wit HIV. the study also aims to assess the predictors of quality of life in the same population.

Major Comment

The authors did not desribe whther they are going to adapt the tool or they are jsut translating the 35 item tool into Setswana and use it as it is. Overall, the authors did not describe their methodology fully for example, on assessment of face and content validity, who are the experts who will be involved. the steps for the translation is also not fully described. who are the experts to be involved. When calculating the sample size for the validation process, what was the rationale for using a Cronbach's alpha of 7.0 when the reccommended is 8.0. It will be good to itemise each validation step and then state the corresponding methods e.g. (I) translation (ii) reliability assessment (iii) validity assessment

May you please refer to this document.

https://www.ncbi.nlm.nih.gov/pmc/articles/PMC5463570/#:~:text=Saudi%20J%20Anaesth.,and%20Abdullah%20Sulieman%20Terkawi

7. PLOS authors have the option to publish the peer review history of their article (what does this mean?). If published, this will include your full peer review and any attached files.

Reviewer #1: No

Reviewer #2: No

Reviewer #3: **Yes: **Nyaradzai Arster Katena

---

## [Author Response · Author response to Decision Letter 0]

2 Apr 2024

Journal requirements comments 

Please remove any funding-related text from the manuscript and let us know how you would like to update your Funding Statement. 

 Please amend either the title on the online submission form (via Edit Submission) or the title in the manuscript so that they are identical

Reviewer’s comments 

Reviewer 1

Thank you for submitting your research protocol for your study to PlosONE. Unfortunately, I found this manuscript to not meet the minimum quality standard for a study protocol. I can only advise to

- consider using guidelines, e.g. the COSMIN guidelines for the development and validation of patient-reported outcome measures under www.cosmin.nl

- consider accessing textbooks on the subject matter, whether by the COSMIN group (see "Measurement in Medicine" by Henrica De Vet and co-authors, 2011) or even good review articles in seminal journal (i.e., the Lohr/ISPOR guidelines),

- consider using guidelines by the MAPI trust for the translation of the tool to your language/culture

Just a few pointers:

- Cronbach's alpha is NOT reliability. For reliability, you want to assess inter-rater or test-retest reliability or both. Both should be done using ICCs or kappa coefficients.

- Cronbach's alpha without a hint as to the factor structure of the tool is unintelligible. Also, Cronbach's alpha is not defined for multidimensional scales (see Tenko Raykov's work). When adapting a well-known instrument, you at least want to confirm its factor structure, ideally using measurement invariance techniques.

- Logistic regression analysis in a cross-sectional sample with an ill-specified sampling mechanism (what you do is not "systematic random sampling", by the way: "systematic random sampling" does not exist - please look into any textbook on sampling mechanisms) and formulating this as "predictors" is not an appropriate technique for a validation study. Please reconsider the definition and subtypes of construct validity, see the COSMIN guidelines on how this should be accomplished.

- Presenting a study protocol for a validation study without describing the tool or the validation studies on the tool that have gone before is a feat in itself! I think I have never come across a validation study or protocol that has not described what the tool is actually measuring. You also would want to give a history of its validation journey and - in particular - where you see the main cultural issues when translating into sub-Saharan languages/culture.

- For this part in particular, you would want to consider using cognitive interviewing techniques and most likely quite extensive qualitative techniques to make sure all items, the construct, and the measurement scales are understood the same or need adapting in your culture.

As inspiration, I can highly recommend Eve Namisango's work from Uganda - please have a look into how she accomplished translation work in palliative care

. 

Reviewer 2 

This is a much needed study within the geographical area of Southern Africa where test translation and validation studies are not as prevalent. Much research is needed on psychological measures, as well as the neglected area of the well-being of those infected with HIV.

The student and their promotor should be commended on this exciting proposed study.

The document reads very well, and all the important points and considerations are covered, in my opinion. The projected sample sizes are also good.

All I would suggest is that they add the full stops at the end of sentences (eg. lines 94, 161, 181, 253, 256, and 260). Apart from that I wouldn't change anything, as the protocol is specific enough, but allows adequate room for statistical exploration

Reviewer 3

The authors did not desribe whther they are going to adapt the tool or they are jsut translating the 35 item tool into Setswana and use it as it is. Overall, the authors did not describe their methodology fully for example, on assessment of face and content validity, who are the experts who will be involved. the steps for the translation is also not fully described. who are the experts to be involved. When calculating the sample size for the validation process, what was the rationale for using a Cronbach's alpha of 7.0 when the reccommended is 8.0. It will be good to itemise each validation step and then state the corresponding methods e.g. (I) translation (ii) reliability assessment (iii) validity assessment

Authors' responses

We have diligently formatted the body of the manuscript to adhere as closely as possible to the prescribed format provided by the PLOS One guidelines. We understand the importance of consistency and readability in scientific publications, and we have made every effort to ensure that our manuscript meets these standards

OO’s research protected time was partly supported by the Incentive Funding for Rated Researchers' Grant from National Research Foundation (No:132385). Research reported in this publication was partly supported by the South African Medical Research Council (SAMRC) through its Division of Research Capacity Development under the Research Capacity Development Initiative from funding received from the South African National Treasury. The content and findings reported/illustrated are the sole deduction, view and responsibility of the researcher and do not reflect the official position and sentiments of the funders. 

Thank you for bringing this to our attention. We have rectified the inconsistency between the title on the online submission form and the title in the manuscript to ensure they are now identical. We appreciate your diligence in ensuring the accuracy and coherence of our submission 

Author’s responses

We have considered COSMIN guidelines and it emerged that whereas COSMIN guidelines provided further elucidation on the distinction between validity and reliability and their measurements, we found a misfit in their relevance to(i) non-clinical trial settings (ii) for an already defined core outcomes set, therefore we incorporated a translation and validation process prescribed by the MAPI trust, as you advised. We also consulted other resources for guidance e.g. [https://www.ncbi.nlm.nih.gov/pmc/articles/PMC5463570/#:~:text=Saudi%20J%20Anaesth.,and%20Abdullah%20Sulieman%20Terkawi]

While we acknowledge your comment on Cronbach’s alpha, we have come across numerous statistical sources that identify Cronbach’s alpha as a measure of internal consistency reliability. Perhaps, we should have been more comprehensive an indicate that we would measure internal consistency reliability using Cronbach’s alpha.[ Here are a few sources to highlight this….. 

[Vaske JJ, Beaman J, Sponarski CC. Rethinking internal consistency in Cronbach's alpha. Leisure sciences. 2017;39(2):163-73.]

Hajjar S. Statistical analysis: Internal-consistency reliability and construct validity. International Journal of Quantitative and Qualitative Research Methods. 2018;6(1):27-38.

We will use “systematic sampling” to comply with your preference however, it is to be recognized that the two terms; systematic random sampling and systematic sampling refer to the same thing and often used interchangeably. In fact, the former is the more complete nomenclature to emphasize that the sampling technique belongs to the probabilistic or random class as opposed to the non-probabilistic/ non-random class. I refer you to the article by Peregrine at Lawrence University and Santa Fe Institute who provides further elucidation on this issue[ https://www.researchgate.net/profile/Peter-Peregrine/publication/247987195_Sampling_in_archaeology/links/5d9606a3299bf1c363f5708b/Sampling-in-archaeology.pdf , manual by Alvi, Mohsin (2016): A Manual for Selecting Sampling Techniques in Research. Herein, we cite a few books and articles in high impact journals that have used “systematic random sampling”. 

Books

1. A Sample of Sampling Definitions Philip Bobko,Shari Miller & Richard Tusing[Pages 157-159 | Published online: 20 Nov 2009

2. (1)

1. Fuller WA. Sampling statistics: John Wiley & Sons; 2011.

Articles

N Engl J Med 1993; 329:661-663/ DOI: 10.1056/NEJM199308263290914

Breast Cancer Research and Treatment 1995 Vol. 36 Issue 1 Pages 55-60/DOI: 10.1007/BF00690185

https://doi.org/10.1007/BF00690185

J Neurosci Methods 2009 Vol. 180 Issue 1 Pages 77-81

Accession Number: 19427532 DOI: 10.1016/j.jneumeth.2009.03.001

https://www.ncbi.nlm.nih.gov/pubmed/19427532

Unless missed, we stated that beyond the validation process, we will apply logistic regression model where upon the HRQoL scores people will be dichotomized into “poor” or otherwise health related quality of life based on the scores and evaluate a set of independent variables as potential predictors of poor HRQoL. To make it clear, validation process, precedes regression modelling and the two are in different phases of the study. 

We have reconditioned the introduction to give a consolidated history of the tool’s validation journey, to capture the pertinent contextual nuances.

Tool Description and Validation History:

• We have included a thorough description of the MOS-HIV tool, its validation journey in various settings, and the cultural considerations relevant to its adaptation to the Setswana language and culture.

• Additionally, we have incorporated insights from Eve Namisango's work in Uganda to inform our approach to translation and validation studies in palliative care.

Statistical Techniques and Sampling Procedures:

• We have addressed concerns regarding statistical techniques by ensuring that appropriate methods are employed for reliability and validity assessments, including inter-rater reliability, test-retest reliability, and confirmatory factor analysis.

• Sampling procedures have been revised to clarify the rationale behind the selection methods and to ensure adequate representation of urban and rural populations.

• 

Thank you very much for your constructive comments and compliments. We have incorporated your suggestions in the revised manuscript 

Thank you for your comments, and we acknowledge the suggestions, and have included more details on measurements on validity and reliability. We believe the details shared are now sufficient for comprehensives and reproducibility. We have further expounded on the translation process-the approach, whom it would involve etc

We plan to adapt, and therefore we will assess the translated tool for its validity and reliability. 

Methodology Clarifications:

• We have provided a more detailed description of the methodology, including the steps involved in the translation process, experts involved in face and content validity assessments, and the rationale behind sample size determination.

• The validation steps have been itemized, and corresponding methods have been stated clearly to enhance transparency and clarity.

Thanks for suggesting this article [ https://www.ncbi.nlm.nih.gov/pmc/articles/PMC5463570/#:~:text=Saudi%20J%20Anaesth.,and%20Abdullah%20Sulieman%20Terkawi ] which we have utilized to guide us through the process of translation and validation process of an existing questionnaire or tool 

Nunnally J. Psychometric Theory. New York: McGraw-Hill; 1978., referenced in the article above indicates that in practice a Cronbach’s alpha of at the least 0.7 has been considered adequate internal consistency.

---

## [Decision Letter · Decision Letter 1]

25 Jul 2024

PONE-D-23-31341R1Evaluating the Validity and Reliability of the Tswana Adaptation of the MOS-HIV Tool for Health-Related Quality of Life among HIV sub-populations in Botswana: A study protocolPLOS ONE

Dear Dr. Molefi,

Thank you for submitting your manuscript to PLOS ONE. After careful consideration, we feel that it has merit but does not fully meet PLOS ONE’s publication criteria as it currently stands. Therefore, we invite you to submit a revised version of the manuscript that addresses the points raised during the review process.

We look forward to receiving your revised manuscript.

Kind regards,

Richard Makurumidze

Academic Editor

PLOS ONE

Journal Requirements:

Reviewers' comments:

Reviewer's Responses to Questions

**Comments to the Author**

1. Does the manuscript provide a valid rationale for the proposed study, with clearly identified and justified research questions?

Reviewer #4: Yes

2. Is the protocol technically sound and planned in a manner that will lead to a meaningful outcome and allow testing the stated hypotheses?

Reviewer #4: Yes

3. Is the methodology feasible and described in sufficient detail to allow the work to be replicable?

Reviewer #4: Yes

4. Have the authors described where all data underlying the findings will be made available when the study is complete?

Reviewer #4: No

5. Is the manuscript presented in an intelligible fashion and written in standard English?

Reviewer #4: Yes

6. Review Comments to the Author

You may also provide optional suggestions and comments to authors that they might find helpful in planning their study.

Reviewer #4: Many thanks to the authors for this well thought out research protocol. This study is very much needed research not only in Botswana but in sub-Saharan Africa, and evidence generated will be useful for policy and practice to improve HRQoL among PLHIV.

Having carefully read the manuscript and rebuttal to reviewers, authors have satisfactorily responded to all concerns raised in the initial review. However, I will suggest that authors check:

Line 89: a bracket without content, probably authors missed a citation.

Line 85-87: In addition to the citation, I will suggest that authors include the MOS-HIV tool as a supplementary file for easy referencing by readers

Line 130: ….’’Understanding HRQoL changes during treatment, particularly for conditions like 130 cryptococcal meningitis, is crucial for improving patient care and outcomes’’ authors should consider inserting a citation to support this statement

Study Design and settings

- Study location: where is Princess Marina Hospital located in Botswana? Include the name of town and county or state

7. PLOS authors have the option to publish the peer review history of their article (what does this mean?). If published, this will include your full peer review and any attached files.

Reviewer #4: No

---

## [Author Response · Author response to Decision Letter 1]

9 Sep 2024

We have uploaded a detailed rebuttal letter addressing each of the reviewers’ comments point by point, along with a revised document with tracked changes and the final version labeled ‘manuscript__0824.

---

## [Editor Report · Decision Letter 2]

27 Sep 2024

Evaluating the Validity and Reliability of the Tswana Adaptation of the MOS-HIV Tool for Health-Related Quality of Life among HIV sub-populations in Botswana: A study protocol

PONE-D-23-31341R2

Dear Dr. Molefi,

We’re pleased to inform you that your manuscript has been judged scientifically suitable for publication and will be formally accepted for publication once it meets all outstanding technical requirements.

Kind regards,

Richard Makurumidze

Academic Editor

PLOS ONE
---

## [Editor Report · Acceptance letter]

2 Oct 2024

PONE-D-23-31341R2 

PLOS ONE

Dear Dr. Molefi, 

I'm pleased to inform you that your manuscript has been deemed suitable for publication in PLOS ONE. Congratulations! Your manuscript is now being handed over to our production team.

Kind regards, 

on behalf of

Dr. Richard Makurumidze 

Academic Editor

PLOS ONE